# Federated Continual Learning with Differentially Private Data Sharing

**Giulio Zizzo**
IBM Research Europe
giulio.zizzo2@ibm.com

**Ambrish Rawat**
IBM Research Europe
ambrish.rawat@ie.ibm.com

**Naoise Holohan**
IBM Research Europe
naoise@ibm.com

**Seshu Tirupathi**
IBM Research Europe
seshutir@ie.ibm.com

## Abstract

In Federated Learning (FL) many types of skews can occur, including uneven class distributions, or varying client participation. In addition, new tasks and data modalities can be encountered as time passes, which leads us to the problem domain of Federated Continual Learning (FCL).

In this work we study how we can adapt some of the simplest, but often most effective, Continual Learning approaches based on replay to FL. We focus on temporal shifts in client behaviour, and show that direct application of replay methods leads to poor results. To address these shortcomings, we explore data sharing between clients employing differential privacy. This alleviates the shortcomings in current baselines, resulting in performance gains in a wide range of cases, with our method achieving maximum gains of 49%.

## 1 Introduction

The interaction of Federated Learning (FL) and Continual Learning (CL) is a underexplored area. CL focuses on training a model when the underlying data distribution changes in time. The trained model needs to perform well on *all* previously seen data modalities, despite only having access to the most recent data distribution. Default training of neural networks in CL leads to *catastrophic forgetting*, where old data modalities are forgotten as the neural network trains on new tasks.

A range of research has emerged to tackle this problem. The developed methods can be broadly categorised into replay [1], regularisation [2], or parameter isolation [3] techniques. Simple replay strategies can involve storing data to form a replay buffer. Despite replay buffer methods often having relative simplicity, they can achieve very strong performance [4]. It is this strategy of replay methods which we develop for FL in this paper.

Performing Federated Continual Learning (FCL) introduces new challenges: different clients can experience different tasks, or clients join and leave the FL system over time causing valuable information only held by certain clients being lost. However, replay methods can be limited in their impact in FL with the default inability to share replay buffers across clients due to privacy concerns.

We focus on the challenge of intermittent client participation in FCL. We tackle the scenario where as time passes clients dynamically join a FCL system, bringing with them current task data. However, clients which dynamically join do not have access to data belonging to tasks which occurred prior to them joining. We model client participation with a FL system as decaying with time. This causes problems, as data pertaining to old tasks is sampled infrequently causing catastrophic forgetting.

Workshop on Federated Learning: Recent Advances and New Challenges, in Conjunction with NeurIPS 2022 (FL-NeurIPS'22). This workshop does not have official proceedings and this paper is non-archival.

We propose a privacy-performance trade-off where users share differentially private (DP) data statistics with each other, enabling access to a common pool of data containing information relating to prior tasks which provides strong performance boosts. The contributions of this paper are as follows:

- We examine a novel scenario in FCL which considers clients dynamically joining the FL system, and having varying participation rates in training rounds. This is a challenging situation, as it can lead to clients which have valuable information on old tasks communicating updates very infrequently. To the author's knowledge this has not been previously studied.

- We propose extending replay methods from using only a client's local data, to additionally utilising shared data between clients with DP noise to preserve privacy. We perform empirical evaluations and show how performance varies with differing levels of DP noise and client participation.

## 2  Background

### 2.1  Continual Learning

Continual Learning focuses on the scenario where we wish to learn from a continuous stream of data, gradually incorporating new knowledge into a model. The core challenge arises from the requirement of only having the current data available for training by default. This causes the phenomenon of catastrophic forgetting where performance on old tasks degrades over time as new tasks are added.

There are a wide range of setups that can be investigated in CL with three popular scenarios being [5]:

- Task Incremental: In this case, data is presented in a series of tasks that the model needs to solve, and the task-ID is provided both at training *and* inference time.

- Domain Incremental: Here the task-ID is provided at training time. At test time the model needs to correctly classify the data, but does not need to infer which task it belonged to.

- Class Incremental: The final scenario is when the model needs to both correctly classify data at test time *and* correctly identify the task. The task-ID is provided only during training.

To handle these problems many strategies have been proposed which can be grouped under three camps: replay, regularisation [2, 6] or parameter isolation [3, 7]. Most relevant to this work are replay methods. They either store raw samples or generate pseudo-samples. In iCarL [1] class exemplars are stored to best compute class means in feature space. Alternatively, [4] showed a simple yet robust baseline of training on old samples yields surprisingly high performance, often beating significantly more complex methods. Lastly, training on GAN generated data has been examined [8]. However, the compute requirements can be high with multiple generative models needed for CL scenarios.

### 2.2  Federated Continual Learning

Research has begun tackling the challenges of performing FCL [9]. Recent work in [10] examined decomposing neural network weights into global and task-specific parameters. Each client then adaptively weights contributions from other clients by taking a weighted combination of their task-specific parameters. The work of [11] used a federated version of prototypical networks to address continual learning problems. In [12] the focus was do detect distribution shifts in a FCL setting, as ill defined task-IDs presents a fresh set of challenges if they are not well specified at training time.

The strategy we propose of privatised data sharing can be used standalone, or employed orthogonally to augment many of the above strategies. This presents a privacy-performance trade-off, however we argue that with many works breaking the privacy of FL, privatised data sharing should not be considered prohibited in FL. Here, we continue to exploit the decentralised nature of training and use differential privacy to provide a base level of protection to any shared data.

### 2.3  Differential Privacy

Differential privacy (DP) has emerged as a popular method to preserve the privacy of sensitive datasets [13]. Through the addition of random noise, DP prevents information leakage about any individual example. For the purpose of this paper, we consider the classical *pure DP*, parametrised by $\epsilon$. Here, a DP mechanism $M$ with domain $\mathcal{X}$ and range $\mathcal{R}$ is $\epsilon$-DP, if for all measurable sets $\mathcal{S} \subseteq \mathcal{R}$, and for any

two adjacent databases $D$ and $D'$, the following is satisfied: $\Pr[M(D) \in \mathcal{S}] \leq e^{\epsilon}\Pr[M(D') \in \mathcal{S}]$. To achieve $\epsilon$-DP in this work we utilise the Laplace distribution [13].

# 3 Proposed Algorithm

## 3.1 Problem Definition

We now more precisely define the problem setup. The classical scenario in CL is a series of tasks $\{\mathcal{T}^{(0)}, \mathcal{T}^{(1)}, \ldots, \mathcal{T}^{(N)}\}$ in which $\mathcal{T}^{(i)}$ is a dataset comprised of $K$ data-label pairs $(x, y)$. The model has access to the data belonging to the current task $\mathcal{T}^{(i)}$, but no other tasks. The goal of our learner is to minimise the loss incurred on the current and all prior tasks. However, note we can only optimise the model using the *current* task data $\mathcal{T}^{(i)}$. Once training is finished on $\mathcal{T}^{(i)}$ and we move to task $\mathcal{T}^{(i+1)}$ the data from $\mathcal{T}^{(i)}$ is inaccessible.

Training is conducted in a FL manner and each task $\mathcal{T}^{(i)}$ brings in new clients $\mathcal{C}^{(i)}$ which have data belonging to $\mathcal{T}^{(i)}$, but they do *not* have data from prior tasks. Clients which were already present in the system at prior tasks $\{\mathcal{C}^{(0)}, \mathcal{C}^{(1)}, \ldots, \mathcal{C}^{(i-1)}\}$ have their own local data shift to $\mathcal{T}^{(i)}$. Hence, if a client joined in task $\mathcal{T}^{(i-1)}$, on a task change, would have their current data update to follow $\mathcal{T}^{(i)}$. For example, they may encounter new classes in $\mathcal{T}^{(i)}$ and loose access to the classes in $\mathcal{T}^{(i-1)}$. Finally, a client's participation is modelled to be non-constant, and governed by a participation function $f_p(\mathcal{N}_T)$ which is parameterised by the number of tasks a client has participated in thus far $\mathcal{N}_T$.

## 3.2 Algorithm

We tackle this problem by examining privatised data sharing between clients. There is a series of works [14, 15, 16] considering data sharing between clients to improve performance, with [15] coining the phrase "hybrid-FL". The algorithm we propose is based on constructing a replay buffer with old task data which can be sampled from when new tasks occur. Traditionally in FL, a client would apply this using only the locally held data. Instead, here clients form both local replay buffers using locally held data, but also create a global replay buffer which is shared between the participants.

The procedure is simple to implement and execute, and gives strong performance boosts in a range of situations which we will empirically show in Section 5. The algorithm proceeds as follows:

- When a task change occurs from $\mathcal{T}^{(i)}$ to $\mathcal{T}^{(i+1)}$, each client with data belonging to the task $\mathcal{T}^{(i+1)}$ computes a local buffer with $N$ samples comprised of averaging $N$ non-overlapping shards of their current training data on a per-class basis.
- Then the global buffer is constructed in three steps:
    1. Each participating client computes a differentially private mean over their data on a per-class basis. We use Laplace DP noise governed by noise parameter $\epsilon$.
    2. Each client sends the DP class means to the aggregator. The aggregator then re-averages the received data to maintain the required total buffer size producing the global buffer.
    3. The clients then receive the global buffer and train on it in conjunction with the local buffer and current training data.

It is worth highlighting that this method can be trivially used in conjunction with other CL approaches ([2, 17, 18]) as during training time we are simply sampling from an extended dataset.

# 4 Experimental Setup

We experiment over a sequence of 10 tasks. Each task has 150 training rounds before a task change occurs. Each client trains for 1 epoch on their local data in a FL training round. The aggregator averages all the weights received to make the new global model. We use the Permuted MNIST and Fashion MNIST datasets. As the Permuted MNIST dataset does not have spatial correlations we use a 4 layer dense neural network. The exact architecture is given in Appendix A. In all experiments 2 clients are present in the initial task, and every time a task change occurs, 2 new clients join.

As described above, the client replay buffers are composed of a local and global component. For the local component each client generates $N = 5$ means per class present in the current task. The global buffer size is also set to be $N = 5$ means per class.

We investigate when the global buffer is fully distributed amongst all the clients, and compare with the performance obtained if the aggregator does not send global data back to the clients, but rather conducts training itself after aggregation on the replay buffer it holds. In this case the clients will just have the local component of the replay buffer.

For the DP noise we use the Laplace mechanism with differing levels of $\epsilon$ to obtain privacy-performance trade-off results.

We model the client participation as follows:

- We have an initial set of clients present and participating in the FL system. They train and participate in each of the 150 training rounds in the current task.

- Each time a task change occurs new clients join. New clients have training data pertaining to the current task, and when future task changes occur their data distribution will also update. They do not have any data from tasks which occurred before they joined the FL system. Similarly, clients already in the system lose direct access to data from prior tasks following the classical CL setups.

- Clients participate in all of the training rounds for a task in which they joined. However, for subsequent tasks their participation rate drops. The subsequent participation rate is modelled following two distributions. In the first case, the clients follow an exponential decay distribution. Thus, the participation probability on a given FL round of training is given by:

$$
p = \begin{cases} 1, & \text{if } \mathcal{N}_T = 0 \\ \frac{1}{\beta} exp^{-(\mathcal{N}_T/\beta)}, & \text{otherwise} \end{cases} \tag{1}
$$

where $\mathcal{N}_T$ is the number of tasks the client has participated in thus far, and $\beta$ is the scale parameter controlling the participation decay profile.

In the second case, we model the clients as having a fixed uniform participation chance $p$ per training round for tasks $\mathcal{N}_T > 0$.

This presents a challenging scenario in FL, as if data is considered purely on a local level then we will be sampling early task data infrequently as time passes. This means the model will fit poorly to those initial clients, potentially resulting in them leaving the FL system due to poor model performance.

We examine two styles of CL that can arise in the class-incremental scenario outlined in Section 2.1:

**Novel Inputs Scenario:** In this situation we assume that all the data is randomly divided between clients. The shifts between tasks occurs in the input distribution $x$. We use the Permuted MNIST dataset to generate distinct tasks.

**Novel Inputs & Targets Scenario:** Here we look at when each new task is comprised of a new set of classes, so the shifts between tasks involve both changes in the inputs $x$ and also novel targets $y$. We use two classes per task, and each task selects the next two classes from MNIST starting with the initial task having the first two classes $y = \{0, 1\}$. At the 5th task, once all the MNIST data is used, we then iterate through the Fashion MNIST dataset in the same manner. Thus, in the end, the model needs to perform classifications over 20 classes comprised of both the MNIST (with labels $y = \{0, \ldots, 9\}$) and Fashion MNIST datasets (having labels $y = \{10, \ldots, 19\}$). On each new task, $\mathcal{T}^{(i)}$, the new task data is divided randomly between the clients present in the task $\{\mathcal{C}^{(0)}, \ldots, \mathcal{C}^{(i)}\}$.

**Baseline Setup**: For our baseline we compare with the setup in which the replay data buffer is kept purely locally to the client. The buffer is computed in the same manner via computing per class averages on the local data. As data is not shared, no DP operations are applied. We use $N = 10$ averaged samples per class for the baseline configuration.

## 5   Results

We run the described experiments, repeating each configuration 3 times reporting the mean result for both types of client participation. The results are shown in Table 1 highlighting the privacy-

| Performance Difference Compared to Baseline | | | | | | |
|---|---|---|---|---|---|---|
| | | $\epsilon$ | Exponential | | Uniform | |
| | | | Client Training | Aggregator Training | Client Training | Aggregator Training |
| Novel Inputs | Low Client Participation | 1.0 | 16.28 | 11.81 | 0.78 | 1.38 |
| | | 10.0 | 24.98 | 23.78 | 4.20 | 4.12 |
| | | 100.0 | 25.43 | 26.07 | 6.61 | 1.76 |
| | High Client Participation | 1.0 | 1.90 | 3.16 | 1.84 | 1.35 |
| | | 10.0 | 5.41 | 7.49 | 4.75 | 4.79 |
| | | 100.0 | 4.36 | 4.47 | 5.28 | 2.46 |
| Novel Inputs & Targets | Low Client Participation | 1.0 | 8.035 | 9.018 | 0.08 | 0.13 |
| | | 10.0 | 35.99 | 31.00 | 0.04 | -0.09 |
| | | 100.0 | 49.06 | 47.68 | 0.17 | 0.22 |
| | High Client Participation | 1.0 | 3.34 | 1.15 | 0.15 | 0.13 |
| | | 10.0 | 5.44 | 2.41 | 0.16 | -0.19 |
| | | 100.0 | 6.35 | 5.97 | 0.34 | 0.33 |

Table 1: Results showing the gain or loss in accuracy compared to the baseline of having replay buffers kept purely locally (higher is better). When considering exponential client modelling we show the performance at different $\epsilon$ levels with both the fastest rate of client decay with $\beta = 0.5$, corresponding to a low level of client participation, as well as the slowest rate of participation decay, $\beta = 2.0$, corresponding to high levels of client participation. Similarly, we show results for uniform client modelling with the highest ($p = 0.5$) and lowest ($p = 0.05$) rates of client participation. This is for both the configurations studied: with either the aggregator holding the global replay buffer (*Aggregator Training*), or it being distributed to all the clients (*Client Training*).

performance trade-offs at the different limits of client participation investigated. Intermediate levels of participation interpolate between these results and values.

The key result is that the lower the client participation, the more pronounced the effects of having a shared global buffer become. For example, for uniform client participation, the joining probabilities were always high enough such that gains are modest when using a global buffer - hence the maximum gain in performance was just 6.61%. However, with exponential client participation, the joining probability can fall very rapidly and thus it becomes crucial to have a distributed global buffer. There are therefore large gains in performance with the largest boost being 49.06% due to the addition of a global buffer.

Hence, the proposed algorithm should be applied in a context dependant manner: if in FCL clients are expected to participate with profiles similar to the exponential decay scenario, then deploying a global buffer in the manner described would alleviate many of the underlying issiues. However, if clients are participating frequently enough such that old task data is sampled sufficiently from the local replay buffer (such as when we examined a uniform participation profile) then using a global buffer will only give small benefits which may not outweigh the additional communication and privacy costs.

Regarding the effects of DP, generally, lowering the level of DP noise results in the accuracy correspondingly rising, particularly with exponential client modelling. However, there are a number of experimental configurations which had little performance difference between $\epsilon = 10.0$ and $\epsilon = 100.0$ illustrating that the increased privacy does not always need to result in a performance drop. To gain a better intuition as to what a client's shared data looks like, we show examples from the datasets in the Appndix.

In almost all cases, training with a decentralised global buffer, compared to a global buffer only held by the aggregator, performed in a similar manner. Hence, the decision of whether to distribute the global buffer does not have to be influenced by the performance, but can be taken based on factors such as the privacy tolerance of the clients, and the compute resources available to the aggregator.

In Table 2 we show the case in which clients participate in every round representing the upper limit of client participation. In that setup sharing data is of limited benefit, so the performance is similar to training on purely local data. We did have an unexpected result when clients always participated for the *Novel Inputs* scenario. At the lowest levels of DP noise with $\epsilon = 100.0$ over-fitting occurs, with

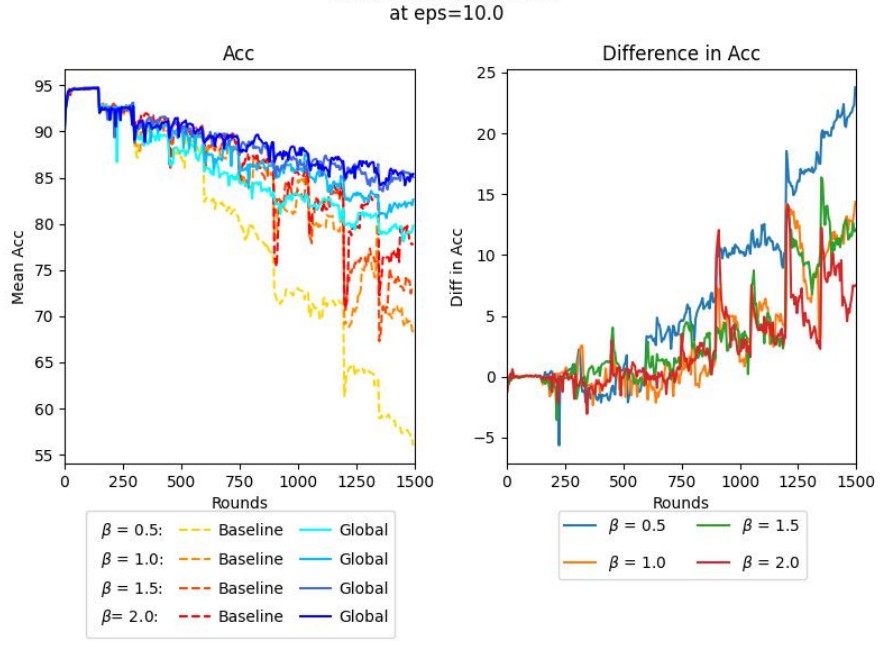

(a) Performance when the global buffer is distributed between clients. (Left) The raw test accuracy across all tasks as a function of the FL training rounds. (Right) The difference in accuracy between the baseline and the global data buffer setups.

(b) Performance when the global buffer is not distributed to the clients and instead used for training at the aggregator level. (Left) The raw test accuracy across all tasks as a function of the FL training rounds. (Right) The difference in accuracy between the baseline and the global data buffer setups.

Figure 1: Results for the *Novel Inputs* task types when clients are modelled with a exponential participation rate at $\epsilon = 10.0$

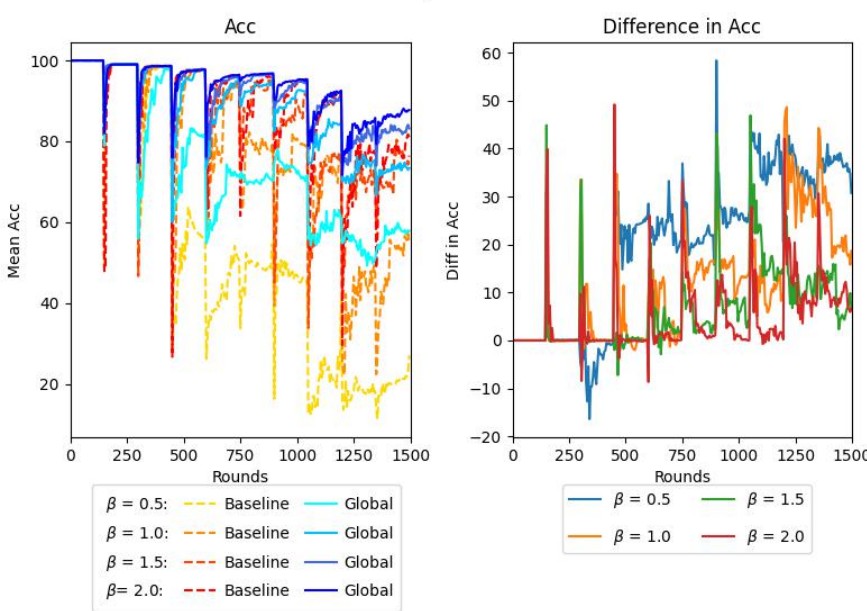

(a) Performance when the global buffer is distributed between clients. (Left) The raw test accuracy across all tasks as a function of the FL training rounds. (Right) The difference in accuracy between the baseline and the global data buffer setups.

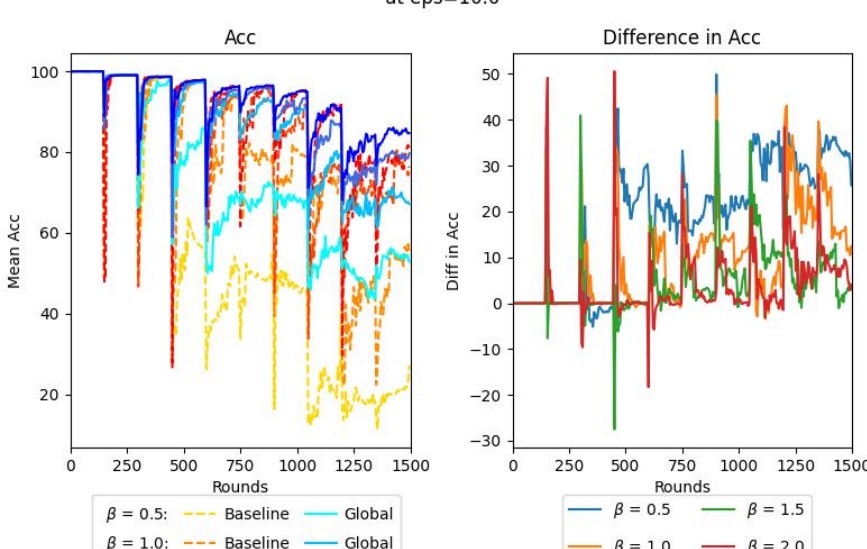

(b) Performance when the global buffer is not distributed and instead used for training at the aggregator level. (Left) The raw test accuracy across all tasks as a function of the FL training rounds. (Right) The difference in accuracy between the baseline and the global data buffer setups.

Figure 2: Results for the *Novel Inputs & Targets* task when clients have a exponential participation rate at $\epsilon = 10.0$.

| Ideal Client Participation | | | |
| --- | --- | --- | --- |
| | Epsilon | Client Training | Aggregator Training |
| Novel Inputs | 1.0 | 2.09 | 1.07 |
| | 10.0. | 4.52 | 4.99 |
| | 100.0 | 1.39 | 0.002 |
| Novel Inputs & Targets | 1.0 | -0.07 | 0.17 |
| | 10.0 | 0.24 | -0.07 |
| | 100.0 | 0.18 | 0.08 |

Table 2: Results showing the gain or loss in accuracy compared to the baseline of having replay buffers kept purely locally (higher is better). Here we report results when clients participate in every round as an idealized case.

the low noise configurations performing worse than those with higher levels of noise, although still performing comparable to, or slightly better than, the baseline accuracy.

We show the full performance curves during training at various levels of client participation in Figures 1 and 2 at $\epsilon = 10.0$. For space constraints we limit ourselves to these configurations to fully plot out. We show both the raw accuracy of our method along with the baseline and highlight the performance difference. We can see in this case that our method outperforms the baseline across the range of $\beta$ values we investigated (0.5 - 2.0).

# 6 Conclusion

In this work we have examined how replay based methods can be extended to work in a FCL setting by sharing data characteristics with DP noise. Here we have focused on temporal variation in client participation, with clients participating less in the training process as time passes. We saw in this case that even with relatively high levels of DP noise we often have performance gains due to the lack of participation from clients who hold valuable data.

# Acknowledgments

This work has been partially supported by the MORE project (grant agreement 957345), funded by the EU Horizon 2020 program.

## A  Appendix: Model Architecture

Neural network architecture used for our experiments. FC $N$ indicates a fully connected layer with $N$ outputs.

FC 1000 → ReLU → FC 1000 → ReLU →
FC 1000 → ReLU →
for *Novel Inputs*: FC 10
for *Novel Inputs & Targets*: FC 20

## B  Appendix: Example Data

We show example data from the MNIST and Fashion MNIST datasets with differing $\epsilon$ noise budgets.



$\epsilon = 1.0$    $\epsilon = 10.0$    $\epsilon = 100.0$

Figure 3: Example data with different levels of $\epsilon$ for the MNIST (top row) and Fashion MNIST (bottom row) datasets.

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
