# OpenReview forum: "Federated Continual Learning with Differentially Private Data Sharing"
_NeurIPS.cc/2022/Workshop/Federated_Learning — FL-NeurIPS 2022 Poster_

### Official Review · Reviewer_HwbZ · 2022-10-12
**Clear, Simple Approach but Results Lacking in Clarity**

The paper presents an approach to tackling continual learning problems in a federated learning context, creating global replay buffers for different tasks to enable continual training on all relevant data and to mitigate catastrophic forgetting. The results show that elements of the proposed system are useful and others are not as useful, and a sweep over relevant parameters is performed to highlight relevant settings.

Strengths:
* The paper presents a clear overview of the continual learning problem as it relates to federated learning, including the temporal nature of the setup.
* The proposed approach is very simple, clear, and easy to follow.
* The experiments are conducted over several hyper-parameter settings, showing the effects of different settings on final performance.
* The dataset used features diverse objects, merging Fashion MNIST and MNIST together.
* Elements of the system achieve high performance relative to naive federated averaging.

Weaknesses:
* The presentation of results is confusing, using relative performance rather than actual final performance numbers. It isn't clear if higher or lower is better, as the text indicates that the bottom row of Table 1 is an upper-bound on performance (Line 186), though reading a 49% change in relative performance seems to suggest a large positive boost to accuracy?
* Testing several hyperparameters and data-bucketing strategies is informative, but ultimately the results come across more as a "throw everything at the wall and see what sticks" presentation. It isn't clear _why_ certain strategies are better than others. For example, the global buffer does not seem to improve performance, so it is not clear why it is being included as part of the core method/algorithmic contribution.
* Throughout the results, numbers and figures are presented to the reader but without description of what they mean or why they are significant.
* Figures 1 & 2 are very difficult to interpret, with small fonts and only color to distinguish different methods/results. Focusing on one of these and showing a larger, clearer picture (with more description of what the reader should be taking away) would be more informative.

The first portion of the paper is clear and easy enough to follow, but the results section needs more attention to present actionable takeaways for readers or useful lessons for the problem setup or approach. As it stands currently, there is not enough description of why certain design choices were made, how the results relate to each other, or which aspects of the experiments/approach were significant (or why).

---

### Official Review · Reviewer_PVdS · 2022-10-16

The authors propose a replay-based continual learning approach in the federated learning context. Results show that data sharing with differential privacy boosts the performance compared to the direct application of replay methods, especially in the exponential participation scenarios. However, the work lacks the analysis of the impact of the proposed method on communication cost.

---

### Official Review · Reviewer_3Tth · 2022-10-18
**weak accept**

This paper studies the application Continual Learning approaches based on replay to FL and show that direct application of these approaches lead to poor performance. The authors then purpose mechanisms of differentially-private data sharing between clients which outperforms the baselines.

Pros:
The intersection of federated learning and continual learning remains relatively unexplored, and I believe that this work would be of interest and invoke future works at the workshop.

Cons:
The theoretical justification or analysis on the propose algorithm is limited, and the experiments only used the MNIST dataset.
Overall:
I think this work opens up an interesting topic on how to do continual learning in the federated setting, but more comprehensive investigations are needed both from the experimental and theoretical perspective.

---

### Decision · Program_Chairs · 2022-10-20

Accept (Poster)